

# Improved GNSS-R bi-static altimetry and independent DEMs of Greenland and Antarctica from TechDemoSat-1

Jessica Cartwright[1,2], Christopher J. Banks[3], Meric Srokosz[1]

[1]National Oceanography Centre, Southampton, UK
[2]Ocean and Earth Science, National Oceanography Centre, Southampton, University of Southampton, Southampton, UK
[3]National Oceanography Centre, Liverpool, UK

*Correspondence to*: Jessica Cartwright (jc1n15@noc.soton.ac.uk)

**Abstract.** Improved Digital Elevation Models (DEMs) of the Antarctic and Greenland Ice Sheets are presented, derived from Global Navigation Satellite Systems-Reflectometry (GNSS-R). This builds on a previous study (Cartwright et al., 2018) using

GNSS-R to derive an Antarctic DEM but uses improved processing and an additional 13 months of measurements, totalling 46 months of data from the UK TechDemoSat-1 satellite. A median bias of under 10 m and root-mean-square (RMS) errors of under 53 m for the Antarctic and 166 m for Greenland are obtained, as compared to existing DEMs. The results represent, compared to the earlier study, a halving of the median bias to 9 m, an improvement in coverage of 18%, and a four times higher spatial resolution (now gridded at 25 km). In addition, these are the first published satellite altimetry measurements of

the region surrounding the South Pole. Comparisons south of 88° S yield RMS errors of less than 33 m when compared to NASA's Operation IceBridge measurements. Differences between DEMs are explored and the future potential for ice sheet monitoring by this technique is noted.

## 1  Introduction

The use of reflected L-Band signals from Global Navigation Satellite Systems (GNSS) for Earth observational purposes was first proposed in 1988 (Hall and Cordey, 1988). GNSS-Reflectometry (GNSS-R) is now applied to the characterisation of the Earth's surface predominantly for the monitoring of ocean surface winds (Clarizia and Ruf, 2016; Foti et al., 2015; Foti et al., 2017; Ruf and Balasubramaniam, 2018). It has been investigated for other applications such as altimetry of the ice sheets and

oceans (eg. Cardellach et al. (2004); Cartwright et al. (2018); Clarizia et al. (2016)), soil moisture (Chew et al., 2016) and monitoring of the cryosphere (eg. Belmonte Rivas et al. (2010); Cartwright et al. (2019); Fabra et al. (2012)). The technique has been found to be highly beneficial when applied to the cryosphere not only for sea ice detection (Alonso-Arroyo et al., 2017; Cartwright et al., 2019; Yan and Huang, 2016) and characterisation (Rodriguez-Alvarez et al., 2019) but also for sea ice altimetry (Hu et al., 2017; Li et al., 2017) and glacial ice altimetry (Cartwright et al., 2018; Rius et al., 2017).



The application of GNSS-R to altimetry was originally proposed by Martin-Neira (1993) and has been successfully demonstrated from fixed, airborne and spaceborne platforms. Due to the highly specular nature of reflections from ice-covered surfaces, it is a natural step to apply these techniques to the cryosphere. In these cases, spaceborne platforms have been able to achieve root-mean-square (RMS) errors of below 5 m when applied to limited tracks using the group delay (Hu et al., 2017) and below 5 cm where phase delay is available (Li et al., 2017). As more GNSS-R data have become available from the Low

Earth Orbiter TechDemoSat-1 (TDS-1), it has been possible to use a larger collection of reflections for the construction of Digital Elevation Models (DEMs) of the larger ice sheets, such as Antarctica (Cartwright et al., 2018). The use of GNSS-R offers a unique opportunity to measure the elevation of ice over the South Pole due to the wide variety of incidence angles available through bi-static altimetry enabling this technique to be unrestricted by the orbital constraints of traditional mono-static radar altimetry.

The use of signals of opportunity results in GNSS-R requiring only very low-mass, low-power receiver only systems and is therefore a low-cost method of remote sensing. The approach therefore lends itself to applications in constellation missions in order to increase spatial and temporal resolutions. Cyclone GNSS (CYGNSS) was launched in 2016 by NASA for the monitoring of winds inside tropical cyclones, and has an average revisit time of 4 hours (Ruf et al., 2013), however, the low inclination of these satellites (35°) means that their data have little application to the cryosphere. A system similar to that of

CYGNSS, but optimised for cryosphere applications, has been proposed (Cardellach et al., 2018). Currently available spaceborne data over the poles is limited to that of satellite TDS-1, which was placed in a high-inclination orbit (98.4°) and active for a total of 46 months between November 2014 and December 2018. It is these data on which this study is based.

    As stated by Slater et al. (2018), DEMs can help in the understanding of ice sheet hydrology through mass balance calculations, grounding line thickness, and delineation of drainage basins. These further improve understanding of ice dynamics and potential

sea level rise associated with ice sheets. This paper builds upon earlier work done by Cartwright et al. (2018), using an algorithm developed by Clarizia et al. (2016) for the estimation of sea surface height. Here we use improved re-tracking combined with expansion of the GNSS-R dataset and enhanced processing to yield higher accuracies over the Antarctic Ice Sheet. This is then applied to the Greenland Ice Sheet, demonstrating the flexibility of the technique and potential for high resolution observations over these areas. These new DEMs are primarily compared with two high resolution DEMs, exclusively from CryoSat-2 in the

case of the Antarctic Ice Sheet (Slater et al., 2018) and from the European Space Agency Climate Change Initiative's (ESA CCI) composite of CryoSat-2 (Simonsen and Sørensen, 2017) and ArcticDEM (https://www.pgc.umn.edu/data/arcticdem) in the case of Greenland. Brief comparisons are given to two additional DEMs for each ice sheet; these are those by Howat et al. (2014) and Bamber (2001) over Greenland and the Bedmap2 Elevation Data (Fretwell et al., 2013) and Bamber et al. (2009) over Antarctica. Further comparisons are performed over the area south of 88° using the Operation IceBridge elevation dataset.

Cartwright et al. (2018) found this approach gave consistent DEM overestimations in data at higher incidence angles, therefore high incidence angle data (>55°) were discarded. In this study, we remove the incidence angle filter to increase the sample size and add an intermediate processing step, a spatial mean of all points within a certain radius of the point in question. This





accounts for the overestimations of the higher incidence angle data, leading to an overall reduction in error and increase in resolution due to the larger dataset.

This paper will first describe the dataset used and the satellite platform TDS-1 in section 2. Then section 3 will detail the improved methods for height estimation and application over both Antarctica and Greenland. Comparison of the new DEMs against the CryoSat-2 and ESA CCI DEMs are reported in section 4, along with investigations into the areas in which they differ, possible causes of these differences and brief comparisons with other DEMs. Finally, section 5 provides the conclusions of the study.

**2       TechDemoSat-1 and Datasets used**

TDS-1 was launched in 2014 as a technology demonstration platform by Surrey Satellite Technology Ltd. into a quasi sun-synchoronous orbit of 98.4° inclination at an altitude of 635 km. TDS-1 carried eight experimental payloads, one of which was the Space GNSS Receiver Remote Sensing Instrument (SGR-ReSI). It is this sensor from which the data used in this study were acquired. SGR-ReSI is extremely low-mass and -power and constructed from commercial off-the-shelf components. Full

details of the SGR-ReSI can be found in Jales and Unwin (2015). Due to the use of the shared platform in the demonstration operation period (November 2014-July 2017) the SGR-ReSI was only active 2 days in every 8-day cycle whereas it was operating 24-hours in the final phase of the mission (August 2017–December 2018). The instrument could receive up to 4 GPS (Global Positioning System) reflections at any one time. This, combined with the asynchronism of the cycle of TDS-1 with that of the GPS satellites, creates a varying web of specular points over time, increasing the spatial coverage, as well as

providing data over the poles, which has thus far not been possible with standard satellite altimetry due to orbital constraints. Data from TDS-1 are provided as delay-Doppler maps (DDMs) which are maps of the scattered power in the delay and Doppler domains. A smooth reflecting surface results in a strong, coherent signal due to the majority of the power originating from the specular point, with a relatively small glistening zone (Zavorotny and Voronovich, 2000). Such DDMs have a distinct peak in power and very little spreading of the power in the delay or Doppler domain. This is in contrast to rougher reflections (for

example, over the ocean surface) where a pronounced horse-shoe shape is visible due to the spread of the signal in both delay and Doppler caused by signal scatter both in front and behind the specular point. At the wavelength of the GPS signals used (L1-Band, ~19 cm), ice is much smoother than the ocean surface. The strength of this return from ice is ideal for the extraction of height information. DDMs were collected every millisecond and subject to onboard incoherent averaging, producing 1-second DDMs and metadata in 6-hour windows. These data are provided in a publicly accessible database

(www.merrbys.co.uk). Each DDM is composed of 128 delay pixels by 20 Doppler pixels, with respective resolutions of 0.252 chips (0.246 μs) and 500 Hz. The vertical resolution that this produces varies largely depending on the geometries of the GPS satellites and TDS-1 at the time of transmission and receipt.

The data used in this study were taken from the entirety of the TDS-1 mission (November 2014 to December 2018). This incorporates the initial demonstration mission period (until July 2017) and the extension period (October 2017 to end of 2018).

During the extension period, although the SGR-ReSI was in constant operation, it downlinked only data over 0 dB in gain. This



results in a lack of data over the highest latitudes, and produces a bias in sample number over Greenland when compared to Antarctica. Data South of 60° is selected for the Antarctic DEM, and for Greenland data North of 58° N and between -10° to -75° E. The data were filtered following Cartwright et al. (2018), with the exception of the incidence angle filter, as previously detailed. This ensured the elimination of noise, as well as the removal of DDMs where the return lies out of the tracking

window, and those data affected by instrument setting changes.

The most recent version of the CryoSat-2 DEM (Slater et al., 2018) was used as a primary comparison for the Antarctic data, whilst the ESA CCI Greenland Ice Sheet product (hereafter referred to as GL-CCI) was used for validation of the Greenland product. GL-CCI is a composite of ArcticDEM (https://www.pgc.umn.edu/data/arcticdem) and CryoSat-2 measurements (Simonsen and Sørensen, 2017). Two other DEMs for each region have been used for brief comparison, and for full details of

these, readers should see the referenced work. In order to allow comparison of the Antarctic DEM south of 88°S, data from Operation IceBridge have been employed, downloaded from the National Snow and Ice Data Centre (Dataset ID ILATM1B, https://nsidc.org/data/ILATM1B/).

## 3       Improved GNSS-R bi-static altimetry

The algorithm of Clarizia et al. (2016) uses the geometry of the receiver and transmitter satellite locations to estimate the height

of the surface above the reference ellipsoid using the time delay between when the reflected signal is expected (modelled as reflecting off the ellipsoid) and the time of receipt by TDS-1. This delay is estimated from the delay waveform obtained from the DDM at the value of the Doppler that corresponds to the maximum power in the DDM. The waveform is then Fourier transform interpolated such that the sample rate is increased by a factor of 1,000 whilst retaining the original spectrum of the waveform. Previous studies (Cartwright et al., 2018; Clarizia et al., 2016) have used the maximum derivative of the leading

edge of the waveform as outlined by Hajj and Zuffada (2003), however, more recent studies have determined that the leading edge at 70% of the maximum power more directly corresponds to the specular point on the surface (Cardellach et al., 2014; Mashburn et al., 2016). As such it is this delay used in this study ("p70" algorithm), leading to a decrease in error over Antarctica as compared to the original study by Cartwright et al. (2018).

A spatial averaging is applied in order to incorporate higher incidence angle points previously discarded due to the application

of an incidence angle filter. This maintains the quality of the data whilst providing data over the region around the South Pole by taking a mean of all heights within 25 km of each specular point. A mean was used as the simplest approach, with weighted means and median explored, but providing no improvement in accuracy. These spatial averages comprise the scattered data for gridding and comparison of interpolated DEM data. The data were then averaged onto a regular 25 km x 25 km grid. This grid is four times finer (higher resolution) than that used by Cartwright et al. (2018) due to the increase in the number of observations

from incorporating higher incidence angle data and the additional observations from the mission extension of TDS-1. Grid resolutions of 5 km, 10 km and 50 km were also investigated, however 25 km was chosen so as to maximise both the resolution of the DEM and coverage in both hemispheres. This was also used as the radius for the spatial mean described above in order





to ensure consistency. These same methods were then applied to the data over the Greenland study area in order to obtain a DEM of the Greenland Ice Sheet.

Differences were calculated from both the gridded products and the scattered points. For the former, the comparison DEMs are re-gridded to the same grid, before subtracting the comparison data from the TDS-1 estimates. In order to compare the scattered data, the comparison DEMs are interpolated linearly to the locations of the TDS-1 specular points before subtraction from the TDS-1 estimates. Antarctic data is also compared through the use of the IceBridge dataset, whereby the TDS-1 DEM is linearly interpolated to the location of the IceBridge data points.

**Table 1: Comparison of sample numbers and total DEM data coverage (as percentage of glacial ice area with elevation estimates) with different filters and datasets for both Greenland and Antarctica. Heights are calculated using the p70 algorithm and gridded at 25 km.**

| | Antarctica | | Greenland | |
|---|---|---|---|---|
| | n | % coverage | n | % coverage |
| Filters: Cartwright et al. (2018) Dataset: Oct 2014- July 2017 | 1,735,766 | 74.8 | 455,746 | 99.5 |
| Filters: This study Dataset: Oct 2014- July 2017 | 1,954,909 | 90.9 | 540,080 | 99.7 |
| Filters: This study Dataset: Oct 2014- December 2018 | 4,223,821 | 92.8 | 1,050,486 | 99.9 |

Over Antarctica, the methods used here give coverage of an additional 18% of Antarctica's glacial ice area (Table 1) and a
decrease of 45% (9 m) in interpolated median error to 10.4 m, as shown in Table 2, when compared to Cartwright et al. (2018). The RMS error of the DEM (gridded error) shows a decrease of 115 m, as shown in Table 3. This recalculated DEM can be seen in Figure 1. Comparisons of data south of 88°S with available Operation IceBridge (Studinger, 2013) data yields RMS errors of less than 33 m (Table 4).

**Table 2:Comparison of interpolated error using method of Cartwright et al. (2018) and those presented in this study, both for Antarctica and Greenland, applied across the entire dataset of TDS-1, data between October 2014 and December 2018. The TDS-1 Antarctic DEM (top) is compared with CryoSat-2 v1 1 km DEM (Slater et al., 2018), DEM by Bamber et al. (2009) and the surface elevation data from Bedmap-2 (Fretwell et al., 2013). The Greenland DEM (bottom) is compared with the GL-CCI, Bamber (2001) and Howat et al. (2014).**

| | Antarctica | | | |
|---|---|---|---|---|
| | Cartwright et al. (2018) method | This study | | |
| | CryoSat-2 | CryoSat-2 | Bamber DEM | Bedmap-2 |
| Median difference (m) | 19.01 | 10.40 | 10.95 | 10.40 |
| Mean difference (m) | 15.23 | 11.63 | 11.55 | 11.63 |





| | | | | |
|---|---|---|---|---|
| RMS difference (m) | 91 | 52.39 | 56.56 | 52.39 |

| | Greenland | | | |
|---|---|---|---|---|
| | Cartwright et al. (2018) method | This study | | |
| | GL-CCI | GL-CCI | Bamber DEM | Howat DEM |
| Median difference (m) | 17.35 | 9.62 | 48.84 | 26.91 |
| Mean difference (m) | -15.26 | -19.85 | 23.46 | 9.03 |
| RMS difference (m) | 210.15 | 165.73 | 124.24 | 128.88 |

Table 3:Comparison of gridded data between the method of Cartwright et al. (2018) and those presented in this study, both for Antarctica and Greenland, applied across the entire dataset of TDS-1, data between October 2014 and December 2018. The TDS-1 Antarctic DEM (top) is compared with CryoSat-2 v1 1 km DEM (Slater et al., 2018), DEM by Bamber et al. (2009) and the surface elevation data from Bedmap-2 (Fretwell et al., 2013). The Greenland DEM (bottom) is compared with the GL-CCI DEM, that of Bamber (2001) and that of Howat et al. (2014).

| | Antarctica | | | |
|---|---|---|---|---|
| | Cartwright et al. (2018) method | This study | | |
| | CryoSat-2 | CryoSat-2 | Bamber DEM | Bedmap-2 |
| Median difference (m) | -1.20 | 0.40 | 1.05 | 2.98 |
| Mean difference (m) | -67.26 | -24.39 | -13.20 | -13.34 |
| RMS difference (m) | 273.42 | 158.62 | 123.57 | 132.39 |
| | Greenland | | | |
| | Cartwright et al. (2018) method | This study | | |
| | GL-CCI | GL-CCI | Bamber DEM | Howat DEM |
| Median difference (m) | -6.18 | -5.77 | 52.73 | 16.90 |
| Mean difference (m) | -128.39 | -95.88 | -2.31 | -18.96 |
| RMS difference (m) | 322.35 | 274.38 | 215.29 | 205.57 |

Table 4: Comparison of error with Operation IceBridge elevation estimates, (Studinger, 2013). N=2,841,200,289 and N=3,889,345 respectively for continent-wide comparisons and those greater than 88°S

| | Antarctica | |
|---|---|---|
| | whole | >88°S |
| Median difference (m) | 29.27 | -19.55 |
| Mean difference (m) | 15.33 | -15.85 |
| RMS difference (m) | 135.70 | 32.89 |



## 4    Comparison against CryoSat-2 and GL-CCI


As presented in Figure 1, altimetry using GNSS-R is feasible over glacial ice and is capable of giving measurements over the South Pole itself, which is as yet unavailable for measurement with existing satellite altimetry techniques. Interpolated and gridded errors when compared to other DEMs are presented in Table 2 and Table 3 respectively. The DEM product results in a median difference over Antarctica of 40 cm in comparison to the most recent version of the CryoSat-2 DEM and under 6 m

over Greenland when compared to GL-CCI (Table 3). This higher error over Greenland is to be expected considering the higher ratio of steep coastline to inland ice sheet as higher inclinations have been found to be associated with increased error, in agreement with (Cartwright et al., 2018). Data on slope effects can be found in the supplementary information. This is in part due to corner reflection effects giving multiple DDM peaks and to error in the estimation of the specular point location, with surface slope not accounted for in the location calculation. In addition, in Greenland the higher error in these regions may be

due to the high slopes of the coastal terrain resulting in rocky outcrops, rather than glacial ice. In this respect it may be considered similar to the Antarctic Peninsula, and the errors are comparable. These patterns can be seen in Figure 2 with higher errors around the coastlines and in the more mountainous regions of the ice sheet interiors. These points account for the majority of the underestimations appearing near the origin in Figure 3 and are a source of discrepancies between the comparison DEMs themselves, especially where Greenland is concerned. It is these areas that give the large error ranges seen in Figure 2 and 3.


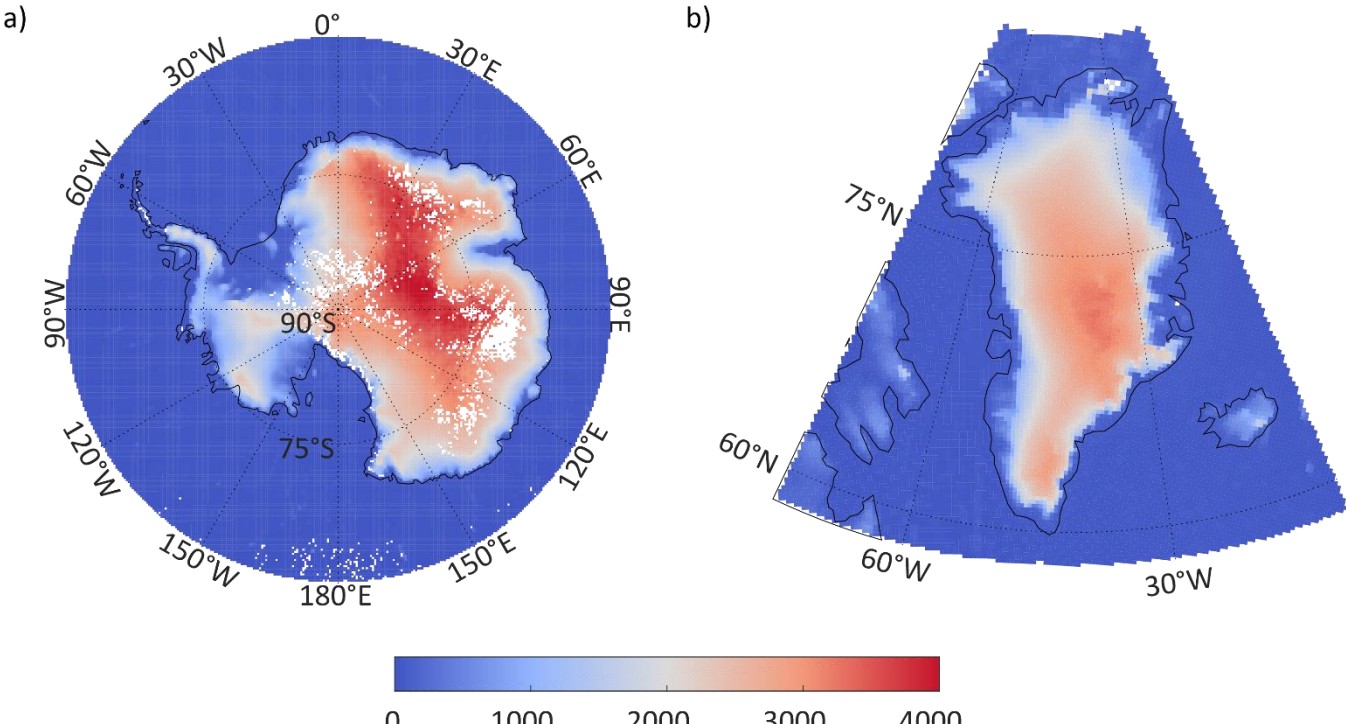

**Figure 1: Digital Elevation models for a) Antarctic and b) Greenland Ice Sheets. Elevations shown are meters above the ellipsoid with white denoting no available data. Gridding in 25 km cells, coastlines black.**



Comparisons with IceBridge data south of 88° were somewhat limited due to the remoteness of the location for surveying. However, the results show RMS errors of less than 33 m. When compared across the full extent of the Antarctic ice sheet, this increases to 136 m, primarily due to the inclusion of steeply sloping ice sheet margins (Table 4).

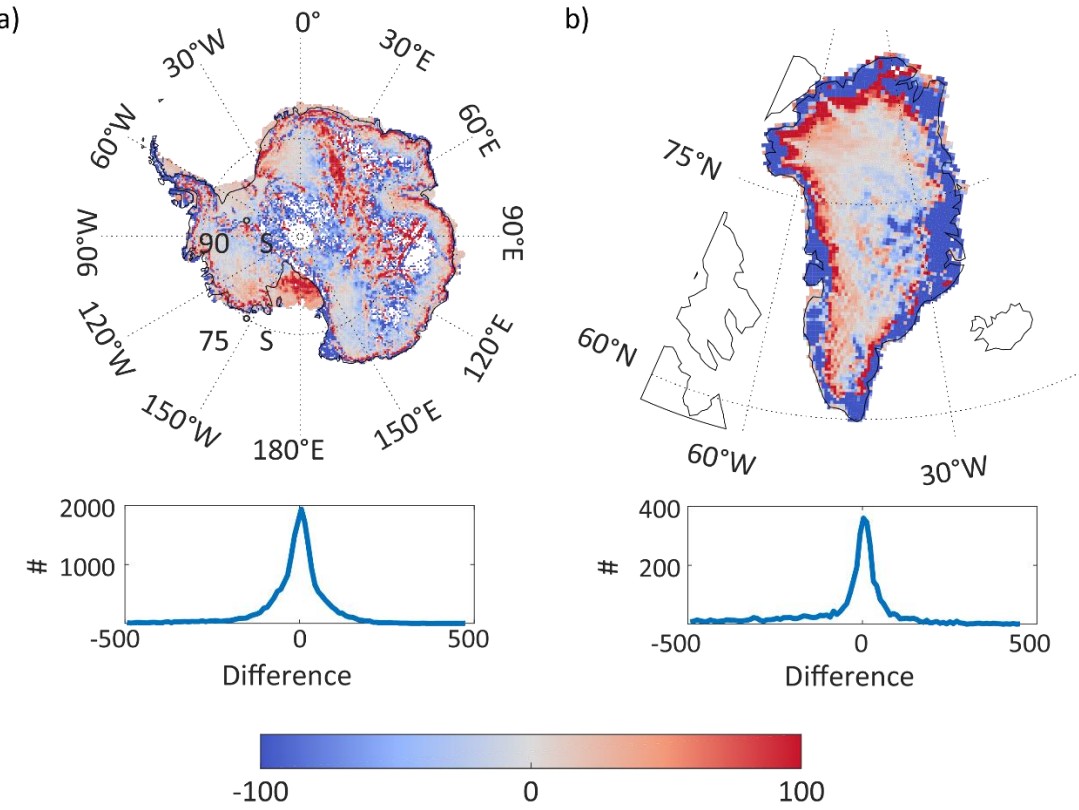

**Figure 2: Error maps over a) Antarctica and b) Greenland with respective histograms (bottom). Error shown is comparison DEM**
**subtracted from TDS-1 DEM .Comparison DEMs are CryoSat-2 v1 1 km DEM (Slater et al., 2018) and GL-CCI for a) and b) respectively.**



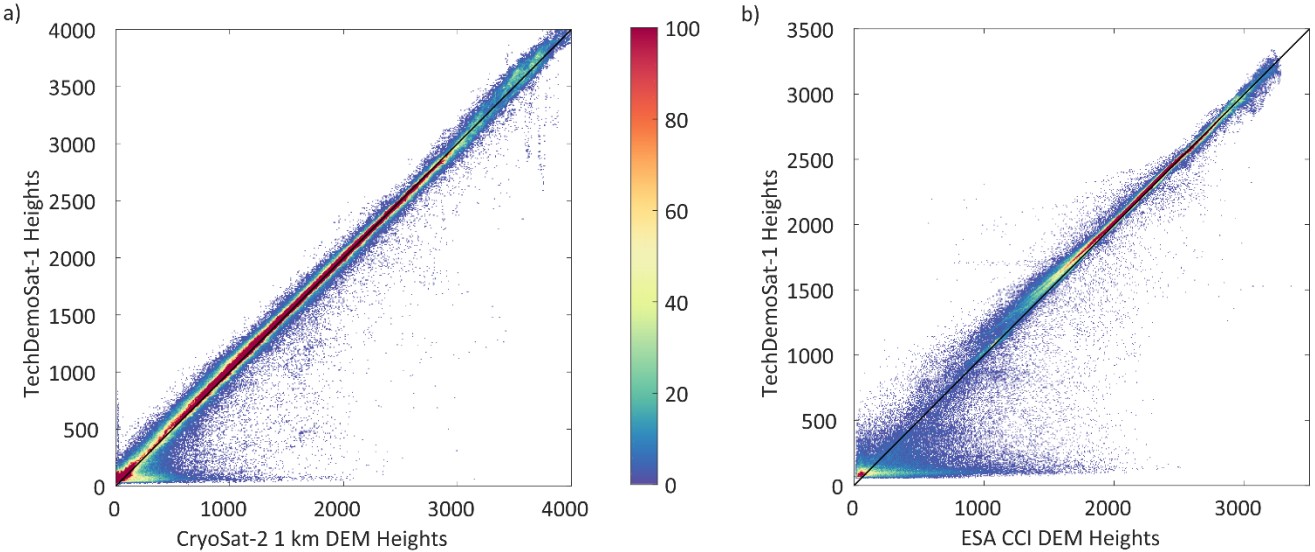

**Figure 3:Density plot comparing height estimations from TechDemoSat-1 over a) Antarctica and b) Greenland and co-located data from the CryoSat-2 v1 1 km DEM (Slater et al., 2018) and GL-CCI respectively. With 1:1 reference line (black)**




When gridded at finer resolutions, accuracy of the resultant DEM increases, however this results in a reduction in the spatial coverage. This suggests that reflections are from a small area and are in agreement with theory that states that the footprint of the SGR-ReSI should be small, at approximately 6 km along-track by 0.4 km across track over sea ice (Alonso-Arroyo et al., 2017). Whilst reflections from glacial ice are expected to be less coherent and therefore produce a larger footprint, it is still

expected to be less than the grid cell size used. Due to the nature of the platform as a demonstration mission, and the design of the system for other measurements, it is necessary to grid the DEMs at this low resolution so as to avoid too many gaps in the data. However, it is promising for future applications of this technology that higher resolution seems to be limited by data availability rather than sensor footprint size.

There are a number of known issues with the TDS-1 data set including the uncertainty of the orbit and attitude of the satellite

itself. These are covered in detail by Foti et al. (2017) and Clarizia and Ruf (2016). Attitude information is acquired from sun sensors, however, when in eclipse this is retrieved from magnetometers with higher uncertainty (at times up to 10°, (Foti et al., 2017)). Large changes in attitude are found in the data when exiting eclipse. However, the error patterns seen here show no obvious relationship to these fluctuations.

The data considered here include those collected in both Automatic Gain Control Mode (November 2014 – April 2015) and

Fixed Gain Mode (April 2015 – end). A strength of the elevation algorithm used here is that it is robust to fluctuations in absolute power levels caused by such changes in mode of acquisition, due to its use of the shape of the waveform and the power relative only to its peak. This is especially valuable as the power received by TDS-1 is uncalibrated with respect to that transmitted from the GPS satellites and not normalised for antenna effects.

Due to the unknown physical properties of the surface, the penetration of L-Band into snow and/or ice is a significant unknown.

This is primarily due to the wide range of electromagnetic changes snow and ice undergo in terms of varying densities and precipitation regimes. Rignot et al. (2001) measured the penetration of L-Band radiation to range between 3 m and 120 m over the Greenland ice sheet, depending on the terrain, with this large range typical of such studies (Li et al., 2017; Mätzler, 2001). An additional factor is the atmospheric uncertainties at high latitudes resulting in ionospheric and tropospheric effects on the signal. These are thought to introduce errors of around 10 m at the equatorial maximum (Hoque and Jakowski, 2012) with

errors being smaller at higher latitudes, and thus these are much smaller than the error magnitudes found here (assuming that the comparison DEMs are "truth", but they too, of course, contribute to the RMS errors).

The different averaging employed for all DEMs produced and used as comparisons here are likely to result in errors when compared to one another. Seasonality and shorter time scale temporal changes were considered, however, they were not found to be connected to the discrepancies between the datasets (results not shown).

**5        Conclusions**

This study demonstrates that high resolution bi-static altimetry of ice sheets is possible with GNSS-R in both hemispheres to an accuracy of under 10 m where compared to contemporary elevation models. With increased data availability through



dedicated GNSS-R missions and sensors designed for the purpose, high resolution altimetry of the polar areas, including the region surrounding the South Pole, would be possible at a higher resolution than that obtained here, where it is limited primarily

by data availability. As the platform requires only a receiver, this technique is inexpensive, lightweight and low power, lending itself to a constellation configuration. Future proposals, such as G-TERN (Cardellach et al., 2018), present the concept of a constellation similar to CYGNSS with a polar focus. Such a mission would allow further increases in the spatio-temporal resolution of the measurements, and through this allow measurements of even the most dynamic aspects of the cryosphere. The feasibility of such a mission would depend on the detailed error budget for the measurements (beyond the scope of this paper).

Accuracies may be increased further through the use of phase delay information (Cardellach et al., 2004; Li et al., 2017) and interferometric techniques. In addition, constraining specular point locations and improved modelling of the signal within the ice sheet will also improve estimates.

## 6        Data availability

Many thanks to the TechDemoSat-1 team at SSTL for making all the collection data publicly available at www.merrbys.co.uk.

Thanks also to the providers of all comparison datasets used here. These are all available publicly. Where the Antarctic DEMs are concerned; these are found for the CryoSat-2 1 km DEM v1.0 at http://www.cpom.ucl.ac.uk/csopr/icesheets2/dems.html; that of Bamber et al. (2009) at http://nsidc.org/data/NSIDC-0422; and the Bedmap2 DEM at https://www.bas.ac.uk/project/bedmap-2. The Greenland elevation models can be found for the ESA CCI product at http://products.esa-icesheets-cci.org/products/details/greenland_digital_elevation_model_v1_0.zip/, that of Bamber (2001)

through the National Snow and Ice Data Centre (NSIDC) at https://nsidc.org/data/nsidc-0092, and that of Howat et al. (2014) at https://nsidc.org/data/nsidc-0645. Operation IceBridge data used for Antarctic comparisons can be found under NSIDC Data Set ID  ILATM1B (https://nsidc.org/data/ILATM1B/).

## 7        Team List

Jessica Cartwright

Dr Christopher J Banks

Professor Meric Srokosz

## 8        Author Contribution

JC, CB and MS designed the study. JC developed the algorithms and analysed the TDS-1 data, and validated the DEM results. JC wrote the paper and JC, CB and MS edited and revised it.



**9      Competing Interests**

The authors declare that they have no conflict of interest.

**10      Acknowledgements**

The authors would like to thank the authors of all comparison data sets as well as the TechDemoSat-1 team at SSTL for making
all their data public access. This work was supported by the Natural Environment Research Council (Grant NE/L002531/1).



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
