# Peer review of "Improved GNSS-R bi-static altimetry and independent DEMs of Greenland and Antarctica from TechDemoSat-1"

_The Cryosphere, 2019_

## Referee Comment (RC1) · Anonymous Referee #1 · 14 Feb 2020

Summary

The authors derived new, static digital elevation models (DEMs) satellite Global Navigation Satellite Systems-Reflectometry (GNSS-R) for both the Greenland and Antarctic Ice Sheets. The DEMs are built off of 46 months of data collected between November 2014 and December 2018 and are posted at 25 km horizontal resolution. The work builds off a previous paper with an improved methodology to incorporate more measurements and shows that the resulting DEMs have better coverage, spatial resolution, and reduced bias in elevation.

Evaluation

[Figure]

Overall, the paper was adequately organized, and there were very few grammatical corrections needed. The methods were laid out sufficiently, although in some cases more detail than a simple citation to the prior paper by Cartwright et al. (2018) would have improved clarity. The largest concern is related to the potential use of the technique for cryospheric purposes. The coarse spatial resolution, coupled with the large uncertainties, makes it challenging to envision a crysopheric research question that would benefit from the technique as presented. The paper would largely benefit from the advice of a glaciologist that is familiar with ice sheet altimetry to strengthen their argument for future potential of ice sheet monitoring with satellite GNSS-R.

A few comments:

1. Line 27 states that the technique is "highly beneficial" to the cryosphere, but without further elaboration, the readers are forced to surmise what applications would benefit. With the results as presented, I was unable to make that connection. More descriptive cryosphere applications would improve the relevance of the paper for the Cryosphere

2. Line 91-92 describes how the vertical resolution varies depending on the satellite geometries, which is completely understandable. However, without any typical range, it really makes it difficult to determine for which applications the technique would be applicable. NASA launched ICESat-2 (laser altimeter) in late 2018, which is capable of monitoring the ice sheets to a precision of 4 mm per year, covering the planet every 91 days. This technique provides a static map and does not have nearly the same precision nor spatial resolution as other existing altimeters (ICESat/ICESat-2/CryoSat-2/etc). Any more insight into the vertical resolution would be largely valuable.

3. Beginning on Line 119, the authors describe the development of the DEM. It would be helpful to know the typical range of measurements that fall within one 25 km grid cell. Or even better, to show a map of the measurement counts within each grid cell for both ice sheets.

4. The Tables and Figures must have the units displayed.

5. Line 169: How much would consideration of slope effects improve the location accuracy of the point location? This seems like an opportunity for improvement of the technique, which could then lend itself to improved potential in future cryospheric applications.

6. Figure 1, please include the units and mask out the regions outside of your DEM (e.g., the southern ocean, etc.)

7. Figure 3, it appears that there are a line of TechDemoSat-2 heights that are biased low against the CryoSat-2 heights near the bottom of both plots. Any idea what this is related, too? Strongly sloping surfaces?

8. As stated by the authors in lines 210-211, there is large uncertainty in L-Band radiation penetration in the snow/firn. Is this fact the reason why the DEM is built over such a large spatial scale? This uncertainty is a very big limitation to the use of this data over the cryosphere where changes at the sub-centimeter scale are quite important.

9. Even with all of the limitations, I was hoping for more discussion of how to best move forward with improving the technique. A completely valid paper on the subject would state all of the limitations (and how we are nowhere near ready to produce numbers that are scientifically useable), but that future improvements will continue to nail down uncertainties and start to answer some of the more relevant concerns regarding the technique. A section at the end describing some of the largest uncertainties in surface elevation retrievals, along with potential future solutions, and what they would mean for the precision of the results would make this paper more relevant for the Cryosphere.

10. In section 6, please state where the TechDemoSat-1 DEMs that were generated are available.

---

## Referee Comment (RC2) · Estel Cardellach (Referee) · 21 Feb 2020

The manuscript is well presented, the study expands and improves a previous one by the same author/s.

A few minor issues:

- What is the difference between Table 2 and 3? (interpolated error and gridded data, what does it mean?). Could these two concepts be clearly explained in the manuscript?

- The Introduction reads: "As stated by Slater et al. (2018), DEMs can help in the understanding of ice sheet hydrology through mass balance calculations, grounding line

thickness, and delineation of drainage basins. These further improve understanding of ice dynamics and potential sea level rise associated with ice sheets.". Which precision is required for DEMs to serve this purpose? Are biases at ten/s of meter level and RMSE at hundred/s meter level sufficiently good for these purposes? (values in Tables 2 and 3).

- Figure 2 and lines 209-212: authors report some relationship between the biases in Figure 2 and topography/terrain slopes. Can they report on potential penetration effects biasing the altimetry over very dry snow –light density of the ice? Cardellach et al., 2012 reported rather deep penetration of GNSS-R signals into Antarctica ice sheet at Dome Concordia. How is this accounted in this study? Only a few sentences are added (page 10, line 209-212), to point that penetration is unknown. However, experimental work with GNSS-R at Concordia Dome (Antarctica) did show large penetration, up to ∼250 m under the very dry/light ice conditions of the area (quite typical of most of Antarctica). Rius et al., 2017 did take penetration into account, reducing the actual geometric path traveled by the signal by considering the slower propagation through dry snow. Would the authors consider a refined DEM with penetration issues accounted for?

---

## Author Comment (AC1) · 17 Apr 2020

The authors would like to thank referee 1 for their comments on the submitted manuscript and believe that by addressing them (see below) the study has been improved and is now suitable for publication in The Cryosphere.

Our response consists of the **reviewer's comments**, our responses and *additions or changes to the text* in that order. These comments are also colour-coded in the attached supplementary PDF.

**Anonymous Referee 1**

**Summary**

**The authors derived new, static digital elevation models (DEMs) satellite Global Navigation Satellite Systems-Reflectometry (GNSS-R) for both the Greenland and Antarctic Ice Sheets. The DEMs are built off of 46 months of data collected between November 2014 and December 2018 and are posted at 25 km horizontal resolution. The work builds off a previous paper with an improved methodology to incorporate more measurements and shows that the resulting DEMs have better coverage, spatial resolution, and reduced bias in elevation.**

**Evaluation**

**Overall, the paper was adequately organized, and there were very few grammatical corrections needed. The methods were laid out sufficiently, although in some cases more detail than a simple citation to the prior paper by Cartwright et al. (2018) would have improved clarity. The largest concern is related to the potential use of the technique for cryospheric purposes. The coarse spatial resolution, coupled with the large uncertainties, makes it challenging to envision a crysopheric research question that would benefit from the technique as presented. The paper would largely benefit from the advice of a glaciologist that is familiar with ice sheet altimetry to strengthen their argument for future potential of ice sheet monitoring with satellite GNSS-R.**

We are pleased the reviewer recognises the organisation and writing quality of the paper. The authors would like to clarify that they do not believe the dataset presented to be adequate in its own right for cryospheric conclusions at present due to the coarse resolution and uncertainties noted in section 4 of the study. However, a system based on these techniques and designed for the purpose would enable the honing of the technique and improvements in error calculations and, therefore accuracy. A further section would be added to the discussion in place of lines 209 – 216 as detailed below, dedicated to the necessary error corrections that would be required in such a system and likely estimates of their magnitude. A full investigation of the error budget and correction of height calculations is outside the scope of the study.

*Addition, replacing lines 209-216*
*5. Discussion of the Benefits and Limitations of the technique*
*In addition to the novelty of measurements over the geographic poles, which were previously not possible with satellite altimetry; the primary benefits of this technique result from the low power and mass of the receiver. These mean that a low cost multi-satellite mission is feasible with the potential to increase the spatial and temporal resolution of observations far beyond those in the present study. The use of a technology demonstration mission limits the data available here and were this technique to be exploited using dedicated platforms designed for these measurements, a significant increase in the available data could be expected. For example, the continuous operation of a single sensor would lead to a 300% increase in data as compared to the initial TDS-1 mission. If, in addition, a larger number of reflections were to be tracked at once, this would also multiply the data available, giving a many-fold increase in the spatio-temporal resolution of products. As seen in this study, the higher resolution of the product gives an increase in accuracy, indicating that the footprint of the measurements is not the limiting factor on the resolution of the data product, but the quantity of data available. This results in a compromise necessary to maximise coverage over the area of interest. A dedicated mission would require a full error budget appraisal, accounting for corrections required due to the design of the sensor and the auxiliary measurements necessary to enable these. It is likely, in addition, that a dedicated mission could also collect phase information from the reflected signals in order to greatly improve the accuracy of the height retrievals, as seen in Hu et al. (2017) and Li et al. (2017).*
*Here we detail sources of error and limitations of this dataset. Due to the unknown physical properties of the surface, the penetration of L-Band into snow and/or ice is a significant unknown. This is primarily due to the wide range of electromagnetic changes snow and ice undergo in terms of varying densities and precipitation regimes as the snow is compacted and the glacial ice formed. Cardellach et al. (2012) measured the penetration of GNSS signals of up to 300 m over dry snow in Antarctica, whilst similar studies at L-Band over glacial ice in Greenland have yielded between 3*

*m and 120 m of penetration depending on the terrain, (Li et al., 2017; Mätzler, 2001; Rignot et al., 2001). These corrections are not applied to the dataset here due to the unknown characteristics of the surface at the time of the retrieval. An additional factor is the atmospheric uncertainties at high latitudes resulting in ionospheric and tropospheric effects on the signal. These are thought to introduce errors of around 10 m at the equatorial maximum (Hoque and Jakowski, 2012) with errors being smaller at higher latitudes, and thus these are much smaller than the error magnitudes found here (assuming that the comparison DEMs are "truth", but they too, of course, contribute to the RMS errors).*

**A few comments:**
**1. Line 27 states that the technique is "highly beneficial" to the cryosphere, but without further elaboration, the readers are forced to surmise what applications would benefit. With the results as presented, I was unable to make that connection. More descriptive cryosphere applications would improve the relevance of the paper for the Cryosphere**
The new section detailed above ("5. Discussion of Benefits and Limitations of the technique") would clarify the advantages of the technique and its potential to increase spatial and temporal resolution of observations over the cryosphere, especially over the regions near the poles where existing techniques have a data "hole".

**2. Line 91-92 describes how the vertical resolution varies depending on the satellite geometries, which is completely understandable. However, without any typical range, it really makes it difficult to determine for which applications the technique would be applicable. NASA launched ICESat-2 (laser altimeter) in late 2018, which is capable of monitoring the ice sheets to a precision of 4 mm per year, covering the planet every 91 days. This technique provides a static map and does not have nearly the same precision nor spatial resolution as other existing altimeters (ICESat/ICESat-2/CryoSat2/etc). Any more insight into the vertical res-**

[Figure]

**olution would be largely valuable.**

As noted on line 90, the delay resolution of the delay-Doppler map is approximately 0.252 chips (0.246 $\mu$s). This equates to approximately 37 m in height of resolution. This, however, is increased through interpolation for the identification of the code delay from the waveform, and future applications of this technique would likely include the collection of phase information, which would yield a further increase in precision of at least an order of magnitude, for example Li et al. (2017) retrieved sea ice heights to an root-mean square difference of 4.7 cm using phase-delay altimetry over select tracks. The below will be added to the manuscript (line 91) to reflect this.

*Addition, line 91, following ". . . Hz"*

*Offering a vertical resolution of 37 m prior to increases in precision through waveform interpolation to 1000 times the resolution.*

**3. Beginning on Line 119, the authors describe the development of the DEM. It would be helpful to know the typical range of measurements that fall within one 25 km grid cell. Or even better, to show a map of the measurement counts within each grid cell for both ice sheets.**

The authors would be happy to add the maps of counts and standard deviation (as below) to the supplementary information on resubmission.

*Addition, Supplementary Information*

*Figure 1 attached: Counts of measurements per 25 km grid cell over Antarctica (left) and Greenland (right).*

*Figure 2 attached: Standard deviation of measurements in metres per 25 km grid cell over Antarctica (left) and Greenland (right).*

**4. The Tables and Figures must have the units displayed.**

The authors note that units were missed from the figures, these will be amended. All elevations are in metres, as specified in the tables and captions.

**5. Line 169: How much would consideration of slope effects improve the location accuracy of the point location? This seems like an opportunity for improvement of the technique, which could then lend itself to improved potential in future cryospheric applications.**

To first order, since height retrieval depends purely on geometry through specular reflection, this should not be affected by the slope of the surface if the specular reflection region is small. The main influence on this in the case of GNSS-R is the size of the glistening zone with respect to the angle and direction of the slope. This is largely dependent on the roughness of the reflecting surface which is itself a large unknown in GNSS-R, especially over the higher relief areas such as the coastal zone of Greenland where it may in fact be rocky outcrops rather than ice sheet. There is certainly much potential to improve this knowledge in further studies, however it is outside the scope of this study.

**6. Figure 1, please include the units and mask out the regions outside of your DEM (e.g., the southern ocean, etc.)**

Figure 1 will be replaced with Figure 3 as attached, with sea surface heights and areas outside the ice sheet of interest masked out.

**7. Figure 3, it appears that there are a line of TechDemoSat-2 heights that are biased low against the CryoSat-2 heights near the bottom of both plots. Any idea what this is related, too? Strongly sloping surfaces?**

Indeed, as stated in lines 169 – 174, it is believed that these underestimates are due to high slopes of coastal terrain and rocky outcrops. We state: "In addition, in Greenland the higher error in these regions may be due to the high slopes of the coastal terrain resulting in rocky outcrops, rather than glacial ice. In this respect it may be considered similar to the Antarctic Peninsula, and the errors are comparable. These patterns can be seen in Figure 2 with higher errors around the coastlines and in the more mountainous regions of the ice sheet interiors. These points account for the majority of the underestimates appearing near the origin in Figure 3 and are a source of discrepancies

between the comparison DEMs themselves, especially where Greenland is concerned. It is these areas that give the large error ranges seen in Figure 2 and 3."

**8. As stated by the authors in lines 210-211, there is large uncertainty in L-Band radiation penetration in the snow/firn. Is this fact the reason why the DEM is built over such a large spatial scale? This uncertainty is a very big limitation to the use of this data over the cryosphere where changes at the sub-centimeter scale are quite important.**

As stated in line 126, the grid at "25 km was chosen so as to maximise both the resolution of the DEM and coverage in both hemispheres". The primary reason it is so coarse is the lack of data, with the demonstration platform operating the SGR-ReSI for 2 days out of every 8. Line 190 goes into more detail on this matter: "When gridded at finer resolutions, accuracy of the resultant DEM increases, however this results in a reduction in the spatial coverage. This suggests that reflections are from a small area and are in agreement with theory that states that the footprint of the SGR-ReSI should be small, at approximately 6 km along-track by 0.4 km across track over sea ice (Alonso-Arroyo et al., 2017). Whilst reflections from glacial ice are expected to be less coherent and therefore produce a larger footprint, it is still expected to be less than the grid cell size used".

The penetration depth is the largest source of uncertainty in the use of these measurements, and will be detailed further with the addition of section 5, as above.

**9. Even with all of the imitations, I was hoping for more discussion of how to best move forward with improving the technique. A completely valid paper on the subject would state all of the limitations (and how we are nowhere near ready to produce numbers that are scientifically useable), but that future improvements will continue to nail down uncertainties and start to answer some of the more relevant concerns regarding the technique. A section at the end describing some of the largest uncertainties in surface elevation retrievals, along with potential future solutions, and what they would mean for the precision of the results would**

**make this paper more relevant for the Cryosphere.**

As suggested by the reviewer, we will add section 5 "Discussion of Benefits and Limitations of the technique" to the study. This is detailed above, and further explores the advantages and error sources in this technique along with the expectations of further missions were they to be designed for this purpose.

**10. In section 6, please state where the TechDemoSat-1 DEMs that were generated are available** The DEMs generated have been submitted to the Polar Data Centre at BAS and the DOIs will be added into the data availability section, as below.

*Addition, Data Availability*

*The DEMs produced in this study are available for download at [doi].*

Cardellach, E., Fabra, F., Rius, A., Pettinato, S., and D'Addio, S.: Characterization of dry-snow sub-structure using GNSS reflected signals, Remote Sensing of Environment, 124, 122-134, 2012.

Hoque, M. M. and Jakowski, N.: Ionospheric propagation effects on GNSS signals and new correction approaches, Global Navigation Satellite Systems: Signal, Theory and Applications, 2012. 381-405, 2012.

Hu, C., Benson, C., Rizos, C., and Qiao, L.: Single-Pass Sub-Meter Space-Based GNSS-R Ice Altimetry: Results From TDS-1, IEEE Journal of Selected Topics in Applied Earth Observations and Remote Sensing, 10, 3782-3788, 2017.

Li, W., Cardellach, E., Fabra, F., Rius, A., Ribó, S., and Martín-Neira, M.: First spaceborne phase altimetry over sea ice using TechDemoSat-1 GNSS-R signals, Geophysical Research Letters, 44, 8369-8376, 2017.

Mätzler, C.: Applications of SMOS over terrestrial ice and snow, 2001, 10-12, 2001.

Rignot, E., Echelmeyer, K., and Krabill, W.: Penetration depth of interferometric synthetic-aperture radar signals in snow and ice, Geophysical Research Letters, 28, 3501-3504, 2001.

Please also note the supplement to this comment:

https://www.the-cryosphere-discuss.net/tc-2019-289/tc-2019-289-AC1-supplement.pdf

[Figure]

a)

b)

1        125        250

**Fig. 1.** Counts of measurements per 25 km grid cell over Antarctica (left) and Greenland (right).

a)

30° W    0°    30° E

60° W

90° W

60° E

90° E

90° S

120° W

75° S

120° E

150° W    150° E

180° W

b)

75° N

60° N

60° W    30° W

0    75    150
Standard Deviation (m)

**Fig. 2.** Standard deviation of measurements in metres per 25 km grid cell over Antarctica (left) and Greenland (right).

a)

b)

30° W

0°

30° E

60° W

60° E

90° W

90° E

120° W

120° E

150° W

150° E

180° E

75° S

90° S

75° N

60° N

60° W

30° W

| | | | | | |
|---|---|---|---|---|---|
| 0 | 1000 | 2000 | 3000 | 4000 | |

Elevation above the ellipsoid (m)

**Fig. 3.** Figure to replace figure 1 in original copy.

---

## Author Comment (AC2) · 17 Apr 2020

The authors would like to thank Estel Cardellach for her comments on the submitted manuscript and believe that by addressing them, as below the study has been improved and is now suitable for publication in The Cryosphere. Our response consists of the **reviewer's comments**, our responses and *additions or changes to the text*. These are also attached as a supplement in a colour-coded format PDF format.

**Estel Cardellach (Referee)**
**estel@ice.cat**
**The manuscript is well presented, the study expands and improves a previous**

**one by the same author/s. A few minor issues:**

**- What is the difference between Table 2 and 3? (interpolated error and gridded data, what does it mean?). Could these two concepts be clearly explained in the manuscript?**

Interpolated error corresponds to the error associated with individual measurements, whereas gridded error reflects the error in the DEM as a gridded product (lines 123 and 130-134). The authors believe that these two different measurements of error shed light on the effect of the gridding process on the accuracy of the measurements as well as comparing the end product more faithfully with other DEMs. The authors will add the below to the text on line 134 to clarify the purposes of the multiple error calculations.

*Addition, line 134*

*. . .*

*TDS-1 estimates. Both types of error (interpolated and gridded) are considered in order to give a comprehensive view of the sources of error both on point measurements and impact on the finished product. Antarctic [. . .]*

**- The Introduction reads: "As stated by Slater et al. (2018), DEMs can help in the understanding of ice sheet hydrology through mass balance calculations, grounding line thickness, and delineation of drainage basins. These further improve understanding of ice dynamics and potential sea level rise associated with ice sheets.". Which precision is required for DEMs to serve this purpose? Are biases at ten/s of meter level and RMSE at hundred/s meter level sufficiently good for these purposes? (values in Tables 2 and 3).**

The authors do not believe the dataset presented to be adequate in its own right for cryospheric conclusions at present due to the coarse resolution and the uncertainties noted in section 4 of the study. However, a system based on these techniques and designed for the purpose would enable the honing of the technique and improvements in error calculations and, therefore accuracy. A further section has been added to clarify

the ways in which a dedicated mission would be expected to improve upon the errors seen here, as well as statements of the magnitude of errors concerned.

*Addition, replacing lines 209-216*

*5. Discussion of the Benefits and Limitations of the technique*

*In addition to the novelty of measurements over the geographic poles, which were previously not possible with satellite altimetry; the primary benefits of this technique result from the low power and mass of the receiver. These mean that a low cost multi-satellite mission is feasible with the potential to increase the spatial and temporal resolution of observations far beyond those in the present study. The use of a technology demonstration mission limits the data available here and were this technique to be exploited using dedicated platforms designed for these measurements, a significant increase in the available data could be expected. For example, the continuous operation of a single sensor would lead to a 300% increase in data as compared to the initial TDS-1 mission. If, in addition, a larger number of reflections were to be tracked at once, this would also multiply the data available, giving a many-fold increase in the spatio-temporal resolution of products. As seen in this study, the higher resolution of the product gives an increase in accuracy, indicating that the footprint of the measurements is not the limiting factor on the resolution of the data product, but the quantity of data available. This results in a compromise necessary to maximise coverage over the area of interest. A dedicated mission would require a full error budget appraisal, accounting for corrections required due to the design of the sensor and the auxiliary measurements necessary to enable these. It is likely, in addition, that a dedicated mission could also collect phase information from the reflected signals in order to greatly improve the accuracy of the height retrievals, as seen in Hu et al. (2017) and Li et al. (2017).*

*Here we detail sources of error and limitations of this dataset. Due to the unknown physical properties of the surface, the penetration of L-Band into snow and/or ice is a significant unknown. This is primarily due to the wide range of electromagnetic changes snow and ice undergo in terms of varying densities and precipitation regimes as the snow is compacted and the glacial ice formed. Cardellach et al. (2012) mea-*

*sured the penetration of GNSS signals of up to 300 m over dry snow in Antarctica, whilst similar studies at L-Band over glacial ice in Greenland have yielded between 3 m and 120 m of penetration depending on the terrain, (Li et al., 2017; Mätzler, 2001; Rignot et al., 2001). These corrections are not applied to the dataset here due to the unknown characteristics of the surface at the time of the retrieval. An additional factor is the atmospheric uncertainties at high latitudes resulting in ionospheric and tropospheric effects on the signal. These are thought to introduce errors of around 10 m at the equatorial maximum (Hoque and Jakowski, 2012) with errors being smaller at higher latitudes, and thus these are much smaller than the error magnitudes found here (assuming that the comparison DEMs are "truth", but they too, of course, contribute to the RMS errors).*

**- Figure 2 and lines 209-212: authors report some relationship between the biases in Figure 2 and topography/terrain slopes. Can they report on potential penetration effects biasing the altimetry over very dry snow –light density of the ice? Cardellach et al., 2012 reported rather deep penetration of GNSS-R signals into Antarctica ice sheet at Dome Concordia. How is this accounted in this study? Only a few sentences are added (page 10, line 209-212), to point that penetration is unknown. However, experimental work with GNSS-R at Concordia Dome (Antarctica) did show large penetration, up to âĹij250 m under the very dry/light ice conditions of the area (quite typical of most of Antarctica). Rius et al., 2017 did take penetration into account, reducing the actual geometric path traveled by the signal by considering the slower propagation through dry snow. Would the authors consider a refined DEM with penetration issues accounted for?** As above, the addition of section 5 ("Discussion of Benefits and Limitations of the technique") before the conclusions will clarify the necessary corrections and error calculations that would be expected were a mission to be dedicated to this technique. The unknown nature of the penetration has been clarified, referring to the unknown conditions of the retrieval, rather than the lack of investigation into the propagation of the signal into various ice and snow types. The varying properties of the snow, firn

and ice are the principal unknowns in this technique, with modelled penetration values ranging from 8 to 120 m in studies by Rignot et al. (2001) over Greenland's glacial ice to 250 m or more in Cardellach et al. (2012)'s for dry snow in Antarctica. As the authors are unable to find reliable data on the characteristics of the snow and ice pack over both ice sheets in order to ensure corrections are as accurate as possible, these corrections are not attempted here but such unknowns are recognised and discussed.

Cardellach, E., Fabra, F., Rius, A., Pettinato, S., and D'Addio, S.: Characterization of dry-snow sub-structure using GNSS reflected signals, Remote Sensing of Environment, 124, 122-134, 2012. Hoque, M. M. and Jakowski, N.: Ionospheric propagation effects on GNSS signals and new correction approaches, Global Navigation Satellite Systems: Signal, Theory and Applications, 2012. 381-405, 2012. Hu, C., Benson, C., Rizos, C., and Qiao, L.: Single-Pass Sub-Meter Space-Based GNSS-R Ice Altimetry: Results From TDS-1, IEEE Journal of Selected Topics in Applied Earth Observations and Remote Sensing, 10, 3782-3788, 2017. Li, W., Cardellach, E., Fabra, F., Rius, A., Ribó, S., and Martín-Neira, M.: First spaceborne phase altimetry over sea ice using TechDemoSat-1 GNSS-R signals, Geophysical Research Letters, 44, 8369-8376, 2017. Mätzler, C.: Applications of SMOS over terrestrial ice and snow, 2001, 10-12, 2001. Rignot, E., Echelmeyer, K., and Krabill, W.: Penetration depth of interferometric synthetic-aperture radar signals in snow and ice, Geophysical Research Letters, 28, 3501-3504, 2001.

Please also note the supplement to this comment:
https://www.the-cryosphere-discuss.net/tc-2019-289/tc-2019-289-AC2-supplement.pdf
* * *

---

## Author Response (AR1)

The authors would like to thank the editor for his comments as below, and believe that by having addressed them in the revised manuscript it is now suitable for publication by The Cryosphere. Please find a list of changes following the point-by-point response to the editor's comments.

5    Comments to the Author:

Dear Dr. Cartwright, dear co-authors,

In my view, the reviewers raised valid points. Note that reviewer 1 has recommended rejection of the manuscript. However, I do not find the review strong enough to justify this decision. The reviewer has mentioned: "The largest concern is related to the potential use of the technique for cryospheric purposes." Clearly, the addition of section 5 to describe more thoroughly the

10    limitations of the method improves your manuscript. Thank you for writing it. Please, make sure to adjust the abstract accordingly.

We thank the editor for recognising the improvements that section 5 makes to the paper, and have added to the abstract that the benefits and limitations of the technique are fully discussed.

I noticed that you have responded to the comments of reviewer 1 but repeatedly failed to indicate how your revised manuscript

15    will be modified to take the comments into account (e.g. comments 1, 5). I would strongly encourage you to improve your manuscript when reviewers felt necessary to raise a point.

The authors would like to clarify that comment number one was responded to through the addition of section 5. Comment number 5 has been addressed through the editing of the below sentence in the discussion of slope effects (line 169):

This is in part due to corner reflection effects giving multiple DDM peaks and to error in the estimation of the specular point

20    location, with surface slope not accounted for in the location calculation, as it is largely dependent of the roughness of the reflecting surface and its alignment with the look angle of the satellite.

Reviewer 1, comment 1: Based on both reviews, I recommend you adjust the term "highly beneficial".

The manuscript has been adjusted to clarify that it is GNSS-R as a whole that is being referred to, rather than this particular application. "Highly beneficial" has been replaced with "effective"

25    Adjusted wording:

GNSS-R has been found to be successful when applied to the cryosphere not only for sea ice detection (Alonso-Arroyo et al., 2017; Cartwright et al., 2019; Yan and Huang, 2016) and characterisation (Rodriguez-Alvarez et al., 2019) but also for sea ice altimetry (Hu et al., 2017; Li et al., 2017) and glacial ice altimetry (Cartwright et al., 2018; Rius et al., 2017).

In the proposed section 5, "Due to the unknown physical properties of the surface, the penetration of L-Band into snow and/or

30    ice is a significant unknown." While surface properties have indeed an effect on the L-band radiation (references of your choice would be welcome), it is my understanding that it really is the properties at depth that influence the most the penetration depth.

https://doi.org/10.1016/j.rse.2017.07.035

https://doi.org/10.1109/TGRS.2014.2312102

We thank the editor for this input and have adjusted the text to reflect that it is the material rather than the surface of the material

35 to which we refer. We have added these references.

In the same section, when addressing the range of penetration depth, I think it is appropriate to reference work on L-band penetration in Antarctica. To my knowledge, this is the most recent study that may be of your interest: https://doi.org/10.3390/rs10121954.

This reference has been added to our existing references on the penetration of L-Band in Antarctica and Greenland to strengthen

40 the paper.

At this stage, please, consider this additional input and submit your revised manuscript with the edits you have already shown in your responses.

Regards

*List of changes:*

[revised manuscript text omitted]